# Percutaneous Coronary Revascularization after Out-of-Hospital Cardiac Arrest: A Review of the Literature and a Case Series

**DOI:** 10.3390/jcm11051395

**Published:** 2022-03-03

**Authors:** Francesca Scavelli, Iside Cartella, Claudio Montalto, Jacopo Andrea Oreglia, Luca Villanova, Laura Garatti, Claudia Colombo, Alice Sacco, Nuccia Morici

**Affiliations:** 1Department of Cardiology and De Gasperis Cardio Center, ASST Grande Ospedale Metropolitano Niguarda, 20162 Milan, Italy; francesca.scavelli@ospedaleniguarda.it (F.S.); iside.cartella@ospedaleniguarda.it (I.C.); claudio.montalto@ospedaleniguarda.it (C.M.); jacopoandrea.oreglia@ospedaleniguarda.it (J.A.O.); luca.villanova@ospedaleniguarda.it (L.V.); laura.garatti@ospedaleniguarda.it (L.G.); claudia.colombo@ospedaleniguarda.it (C.C.); alice.sacco@ospedaleniguarda.it (A.S.); 2School of Medicine and Surgery, University of Milano Bicocca, 20126 Milan, Italy

**Keywords:** out-of-hospital cardiac arrest, percutaneous coronary revascularization, no ST-segment elevation

## Abstract

Out-of-hospital cardiac arrest (OHCA) is still associated with high mortality and severe complications, despite major treatment advances in this field. Ischemic heart disease is a common cause of OHCA, and current guidelines clearly recommend performing immediate coronary angiography (CAG) in patients whose post-resuscitation electrocardiogram shows ST-segment elevation (STE). Contrarily, the optimal approach and the advantage of early revascularization in cases of no STE is less clear, and decisions are often based on the individual experience of the center. Numerous studies have been conducted on this topic and have provided contradictory evidence; however, more recently, results from several randomized clinical trials have suggested that performing early CAG has no impact on overall survival in patients without STE.

## 1. Introduction

Despite advances in percutaneous coronary revascularization and intensive care unit (ICU) management, out-of-hospital cardiac arrest (OHCA) remains associated with high mortality and severe neurological complications [1]. A study reported mortality of approximately 40% among patients who had been successfully resuscitated after OHCA due to ventricular fibrillation (VF) or pulseless ventricular tachycardia (VT) [2].

Even worse outcomes have been associated with OHCA with non-shockable rhythms of presentation [3].

The prognosis of the resuscitated patient after return of spontaneous circulation (ROSC) is conditioned by “post-cardiac arrest syndrome”. This syndrome comprises anoxic brain injury, post-cardiac arrest myocardial dysfunction, systemic ischemia/reperfusion response, and persistent precipitating pathology. The contribution of each of these components in an individual patient depends on various factors, including prearrest comorbidities, duration of the ischemic insult, and cause of the cardiac arrest. Anoxic brain injury is a major cause of morbidity and mortality, and is responsible for approximately two thirds of deaths in the post-cardiac arrest period; the second cause of mortality is represented by progressive haemodynamic deterioration [4].

One important advance in post-ROSC management is the use of therapeutic hypothermia (TH). The introduction of TH consists of the induction and maintenance of a body temperature between 32 and 34 °C for 24 h in patients who remain in a comatose state after ROSC. An Australian study by Bernard et al. and a European study by the Hypothermia After Cardiac Arrest (HACA) working group documented that the use of TH, in patients with OHCA due to ventricular fibrillation, was associated with clinically significant neurological improvement and a 14% improvement in survival 6 months after resuscitation, compared with the control group without hypothermia induction (59% vs. 45%, *p* < 0.02) [5]. However, it is controversial whether TH might trigger infectious complications through a pro-inflammatory effect (including sepsis) or by creating a “sepsis-like” syndrome via an increase in pro-inflammatory cytokines, including interleukin (IL)-1β, IL-8, and tumor necrosis factor (TNF)-α [6].

A significant correlation has been demonstrated between chemokine (C-C motif) ligand 5 (CCL5), also known as regulated on activation, normal T cell expressed and secreted (RANTES), IL-1β level, and mortality risk in patients with acute coronary syndromes [7]. Podolec et al. reported that TH leads to a rapid reduction in CCL5/RANTES levels, which might have a beneficial effect on decreasing the inflammatory response. The same effect was observed for IL-1β during TH. Although CCL5/RANTES levels correlate with cardiac injury and heart failure markers and they decrease during TH, they failed to predict early and late mortality. In contrast, IL-1β level was associated with 180-day survival [8].

There is uncertainty regarding the applicability of TH to patients in cardiac arrest in whom the initial cardiac rhythm is asystole or pulseless electric activity. These patients have significantly poorer outcomes compared with patients with an initial cardiac rhythm of VF/VT.

Recommended postresuscitation care includes, in addition to targeted temperature management, vital-organ support, and treatment of the underlying cause of the arrest. However, the cause of arrest is often unclear immediately after the event, and the lack of a definitive diagnosis can lead to uncertainty regarding the appropriate treatment.

Ischemic heart disease is often associated with OHCA and different studies on patients undergoing coronary angiography (CAG) following a cardiac arrest have reported a variable incidence of coronary artery disease (CAD), acute occlusion, or evidence of plaque rupture. Significant CAD has been described to be present in up to 70% of patients; however, there was an identifiable culprit lesion in approximately 50% of them [9,10,11,12].

The European Society of Cardiology (ESC) and American Heart Association/American College of Cardiology (AHA/ACC) guidelines clearly state that in the case of cardiac arrest and ST-segment elevation (STE) on electrocardiogram (ECG) after ROSC, immediate CAG and revascularization should be performed in order to reduce mortality and improve patient outcomes [13,14]. However, the cause of the arrest is not always obvious, and in the absence of STE the role of immediate CAG is still a matter of debate. Furthermore, the predictive value of the post-resuscitation ECG in diagnosing acute myocardial injury has often been questioned [15]. In different studies, patients without STE following a cardiac arrest had an acute coronary occlusion at the CAG, with reported prevalence varying from 17 to 33% [9,16,17].

## 2. Clinical Studies

Many studies have been published on this topic and have provided conflicting evidence.

Kern et al. reported that early CAG is associated with improved functional outcomes among resuscitated patients of cardiac arrest, regardless of ECG presentation. Nevertheless, similar survival has been observed in patients with and without STE [18]. Improved survival associated with immediate CAG was instead described by Dumas et al. in the case where extracardiac etiology for the arrest is not clearly identifiable, independent of the post-resuscitation ECG pattern. However, their analysis included a significant proportion of patients with STE for whom the benefit of early revascularization was fully recognized [19]. In contrast, Hollenbeck et al. found that a significantly decreased mortality was associated with the use of early CAG, even after excluding patients with STE from their cohort. In their study, coherent with the previously reported data, 27% of patients were found to have an acute coronary occlusion in the absence of STE on ECG [15]. A similar result was described by Elfwén et al. who reported an improvement in short- and long-term survival with a one-year follow-up [20].

These studies have several limitations. The observational design and the absence of randomization may introduce a selection bias, since the indication to perform early CAG may be influenced by the clinical presentation of the patient and the individual judgment of the physician. Moreover, there is a lack of standardized definitions of early and late CAG, and timing to catheterization was variable in the selected patients.

On the contrary, more robust evidence comes from studies suggesting that performing early CAG has no impact on the overall survival in patients without STE.

In the randomized, multi-center COACT trial (Coronary angiography after cardiac arrest without ST-segment elevation), 552 patients with successfully resuscitated OHCA who were unconscious after ROSC, with initial shockable rhythm and no STE at the ECG, were randomized to immediate (median 0.8 h) or delayed (median 119.9 h) CAG (coronary angiography was performed after neurologic recovery). CAG was performed in 97.1% of the immediate angiography group and in 64.9% of the delayed angiography group. At 90 days, 176 of 273 patients (64.5%) in the immediate angiography group and 178 of 265 patients (67.2%) in the delayed angiography group were alive (odds ratio, 0.89; 95% confidence interval (CI), 0.62 to 1.27; *p* = 0.51). No difference was demonstrated in the number of deaths at one year between the two groups and in the remaining secondary end points (survival at 90 days with good cerebral performance or mild or moderate disability, myocardial injury, duration of catecholamine support, markers of shock, recurrence of ventricular tachycardia, duration of mechanical ventilation, major bleeding, occurrence of acute kidney injury, need for renal-replacement therapy, time to target temperature, and neurologic status at discharge from the ICU). In this study, CAD was found in 64.5% of patients who underwent immediate CAG. However, the vast majority of patients had stable coronary artery lesions, and thrombotic occlusions were encountered in only 5.0% of patients. This might explain results, since percutaneous coronary intervention (PCI) is associated with improved outcomes in patients with acute thrombotic coronary occlusion (e.g., in patients with STEMI). Another reason for the lack of benefit of early coronary intervention may be that the majority of non-survivors died of neurologic complications after the cardiac arrest [1,21].

Similar results are provided by the multi-center trial TOMAHAWK (Angiography after out-of-hospital cardiac arrest without ST-segment elevation), conducted on 554 patients ≥30 years of age with ROSC after OHCA and without STE on post-resuscitation ECG [22]. The study randomized patients—regardless of the rhythm of onset (shockable or non-shockable)—with immediate (median 2.9 h) or delayed (median 46.9 h) CAG. In the latter group, CAG was postponed after the first 24 h of hospitalization (except in the case of clinical worsening with electrical instability, cardiogenic shock or the appearance of ST-segment elevation). CAG was performed in 95% of patients randomized to immediate CAG and in 62% of the delayed group. PCI was performed in 37.2% of cases undergoing immediate CAG and in 43.2% of those with delayed CAG. The primary end point was death from any cause at 30 days, which occurred in 54% of patients undergoing urgent CAG and 46% in the control arm (hazard ratio (HR) 1.28; confidence interval (CI) 95% 1.00–1.63; *p* = 0.06). The secondary end point of death from any cause or severe neurologic deficit, occurred more frequently in the immediate CAG arm (64.3% vs. 55.6%; relative risk 1.16, 95% CI 1.00–1.34).

The results of the COACT and TOMAHAWK trial are consistent with the results of several randomized trials that showed no survival benefit of immediate CAG as compared with delayed CAG in patients with myocardial infarction without STE who had not presented with cardiac arrest [23,24,25,26].

Furthermore, the recently published meta-analysis by Verma et al., conducted on 11 studies enrolling 3581 patients, showed no significant difference in mortality at 30 days, neurological outcome, or rate of revascularization between early or delayed CAG among patients with OHCA without STE. Moreover, 30-day mortality was rather related to patients’ comorbidities—namely, diabetes mellitus, chronic renal failure, previous revascularization and lactate level—suggesting that patient selection is crucial to appropriately identify those who would benefit from revascularization. Once more, a limitation of the analysis was that the terminology for early and nonearly CAG was variable in the included studies; moreover, revascularization related complications were not analyzed, limiting discussion about peri-procedural outcomes [27].

Following results from the COACT trial, ESC guidelines of 2020 suggest that, in patients with OHCA and no STE or cardiogenic shock, an unselected immediate invasive strategy (<2 h) is not superior over a delayed one, and that the management of patients needs to be individualized according to their hemodynamic and neurological status [28].

Several randomized clinical trials are still ongoing [29,30,31,32,33]. Results from pilot studies from the DISCO (Direct or subacute coronary angiography in out-of-hospital cardiac arrest), ARREST trial (A randomized trial of expedited transfer to a cardiac arrest center for non-STE ventricular fibrillation OHCA) and from the prematurely terminated PEARL trial (Early coronary angiography versus delayed coronary angiography) are already available and confirmed no significant difference in mortality between early and delayed CAG in OHCA without STE (Table 1).

## 3. Case Series

A 55-year-old man with OHCA associated with shockable heart rhythm that occurred during physical exercise.

The patient had no previous medical history and no family history of cardiac diseases or sudden cardiac death. There was no history of alcohol abuse, recreational drug use or toxic environmental exposure. He had been in good state of health for the entire day, and he was jogging when he suddenly lost consciousness with consequent frontal cranial trauma. Cardiopulmonary resuscitation (CPR) was immediately performed by bystanders and emergency medical service arrived after approximately 10 min. The heart rhythm was identified as shockable with the semi-automatic defibrillator and sinus rhythm was restored after one shock. Emergency endotracheal intubation was performed, and the patient arrived in our emergency room approximately 40 min after the initial arrest.

The ECG after ROSC showed sinus rhythm (Figure 1A), normal atrio-ventricular conduction, and no evidence of ischemic signs. ECG was repeated after positioning of the right precordial leads in the second intercostal space, and did not reveal a Brugada pattern morphology.

Cardiac ultrasound images showed normal size of the left ventricle, and hypokinesia of the mid and apical interventricular septum with mild reduction in ejection fraction that was 45%. There was ectasia of the aortic root and ascending aorta (42 mm), a normal right ventricle, no significant valvular defects, and no pericardial effusion.

Brain computed tomography (CT), performed without administration of intravenous contrast medium, revealed no evidence of infarct, intracranial hemorrhage, or mass lesions.

Diagnostic CAG performed in emergency (Figure 2A) showed two-vessel CAD with chronic total occlusion (CTO) of the left anterior descending artery (LDA), and critical stenosis of the right coronary artery (RCA) and posterolateral artery (PLA) of the circumflex coronary artery. No culprit lesion was identified, and revascularization was not performed at that time.

The patient was transferred to our ICU. Laboratory tests showed normal complete blood count, renal function, thyroid hormones, and serum electrolytes. Creatine phosphokinase levels were within the normal limit and high-sensitive troponin-T was slightly elevated at 35.5 ng/L (0.0–14.0). A mild alteration in hepatic function was detected with a total bilirubin of 1.37 mg/dL (0.25–1.00) and alanine aminotransferase of 87 U/L (3–45). The clinical neurological examination was normal, as well as the brain CT scan performed after 24 h. Extubation was performed after three days due to psychomotor agitation and complete restoration of neurocognitive status was observed after one week. The patient had very high blood pressure values, and hypertensive treatment with combination therapy was initiated.

Revascularization was performed after one week from the cardiac arrest. In the first procedure, PCI with placement of one drug eluting stent (DES) in the RCA and one DES in the PLA was performed. Complete revascularization was obtained after a further 8 days with anterograde recanalization of the CTO of the LDA with PCI and placement of one DES.

An implantable cardioverter device (ICD) was positioned for secondary prevention. Echocardiogram before discharge showed normal cardiac function with no abnormalities in segmental wall motion.

At one-year follow-up, no intervention of the ICD and arrhythmic events had occurred. The patient had a normal echocardiogram and no symptoms of ischemic heart disease.

A 51-year-old man experienced OHCA, not preceded by prodromes, while playing a tennis match.

This gentleman had a past medical history of mild hypertension and no other cardiovascular risk factors, cardiac diseases, or family history of cardiac pathologies. After the cardiac arrest, he immediately received CPR by a bystander, and ROSC was achieved after one shock with the semi-automatic external defibrillator. Emergency medical service arrived in nearly 10 min, the patient was transferred to our hospital after endotracheal intubation, and was placed on mechanical ventilation.

At arrival in our ICU, the ECG showed sinus rhythm, normal atrio-ventricular conduction and ST-segment depression in leads V5–V6. Bedside transthoracic echocardiogram demonstrated a structurally normal heart with preserved left ventricular systolic function; however, hypokinesis of the distal segment of the anterior and lateral wall was evident.

On suspicion of acute coronary syndrome, emergency CAG was performed via radial artery approach (Figure 2B). Angiography revealed a three-vessel CAD with CTO of the LDA in its middle tract, and critical ostial stenosis of the first diagonal branch and of the circumflex coronary artery (LCx). PCI was performed with anterograde recanalization of the CTO of the LDA, and placement of three DESs at the level of the bifurcation with the first diagonal branch.

Blood tests revealed elevated markers of myocardial injury with an hs-Tnt of 412 ng/L (0.0–14.0 ng/L). No other alterations were present. No significant increase in hs-Tnt was observed after the procedure.

Extubation was performed during the second day of stay, after evidence of a preserved neurocognitive status with sedation interruption and a neurological wake-up test. Complete revascularization was achieved after 11 days with PCI and the placement of three DESs in the LCx.

Lastly, cardiac magnetic resonance (CMR) was performed before discharge. Hypokinesia of the anterior and lateral wall with an overall preserved ejection fraction was confirmed. T2-weighted short-tau inversion recovery (T2w-STIR) showed signal hyperintensity at the level of the mid-low segment of the anterior wall, consistent with the presence of edema (Figure 3). Furthermore, late gadolinium enhancement (LGE) with an ischemic pattern was present in the anterior wall and in the apex, with a subendocardial distribution with variable transmural extension, confirming the diagnosis of acute anterior myocardial infarction (Figure 4).

As in the previous case, after one year, the patient was asymptomatic and did not experience any arrhythmic episode.

## 4. Discussion

The aforementioned cases offer different points of discussion. We presented two patients with a similar cardiovascular risk profile and no previous history of cardiac events. Both of them experienced OHCA with a shockable rhythm during physical exercise. However, the first patient had a non-ischemic ECG post-ROSC and a non-significant increase in markers of myocardial injury, while the second patient presented with ischemic ECG changes and contextual increase in troponin that configured the clinical scenario of an acute coronary syndrome although without evidence of an STE.

Clearly, cardiac biomarkers are not immediately available, and the decision to perform emergent CAG and eventual revascularization in these cases is always based on the ECG and clinical evaluation of the patient.

Both patients underwent immediate CAG, yet, according to current guidelines, this decision was definitely questionable, in particular in the first case. No clear culprit lesion was identified and, considering the overall clinical scenario, in the first patient no immediate revascularization was attempted, while in the second, retrograde recanalization of a CTO was performed in emergency. Evidence of at least one CTO in patients with OHCA and no STE is not infrequent and, as reported in recent studies, it is found in approximately one-fourth of cases. Currently, it is not well established whether these CTOs are just bystanders, or if they contribute to the ischemic burden and should be considered for revascularization [18]. Nevertheless, at long-term follow-up, patients with non-revascularized CTOs have a worse outcome compared with those revascularized, with a three-fold risk of sudden cardiac death and ventricular arrythmias [34]. Moreover, in patients with cardiac arrest and acute coronary syndrome, the presence of a CTO is significantly associated with higher in-hospital mortality [35].

With regards to ICD implantation for secondary prevention, ESC and AHA/ACC guidelines recommend it in patients with a documented VF or VT causing haemodynamic instability, in the absence of an identified reversible cause [36,37]. The first patient did not have a diagnosis of an acute coronary syndrome and the cause of the cardiac arrest was not completely clear, so indication to ICD implantation was given for secondary prevention. Contrarily, the second patient experienced an acute coronary event, and even in the absence of a clear culprit lesion, an ischemic pattern with an extent of transmural LGE was evident at the CMR. Considering that complete revascularization was achieved, ICD was not implanted in this case.

## 5. Conclusions

The presented cases demonstrate how, in daily practice, indications to perform immediate CAG are often based on individual decisions, and that clear guidelines are still missing. Many randomized clinical trials are still ongoing, and results are needed to provide a standardized approach to guide patient selection criteria.

According to recent evidence, in patients that survive an OHCA and present with a non-STE post-resuscitation ECG, performing early versus delayed CAG provides no benefits in terms of short-term mortality or neurological outcome. In the resuscitated OHCA, short and long-term prognosis is substantially conditioned by two factors: neurological damage and cardiac damage. An immediate invasive strategy may affect the latter, especially when the onset is characterized by cardiogenic shock or hemodynamic instability (conditions that excluded randomization in TOMAHAWK and COACT, but not enrollment in many registry studies), but it is unlikely to have an effect on the former. Anoxic injury and neurologic impairment were the predominant causes of death in the TOMAHAWK study, as well as in the COACT study. In subjects with significant neurological impairment upon arrival at the hospital (in the TOMAHAWK study the median of the Glasgow Coma Scale score was 3), indicative of a very advanced situation, many invasive procedures could be futile, or even potentially harmful.

## Figures and Tables

**Figure 1 jcm-11-01395-f001:**
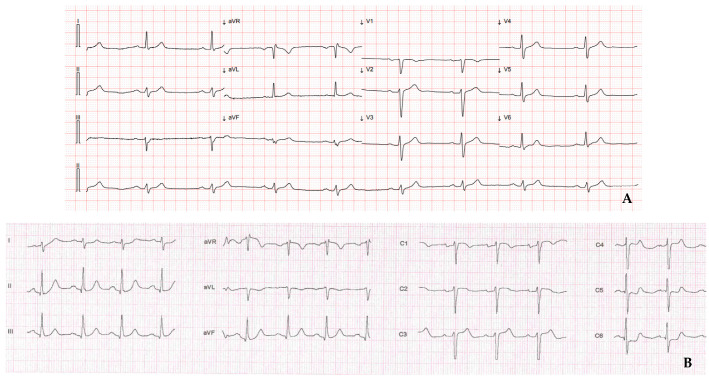
Electrocardiogram after restoration of spontaneous circulation. (**A**) Case 1: no ischemic signs are evident; (**B**) Case 2: ST-segment depression in leads V5–V6.

**Figure 2 jcm-11-01395-f002:**
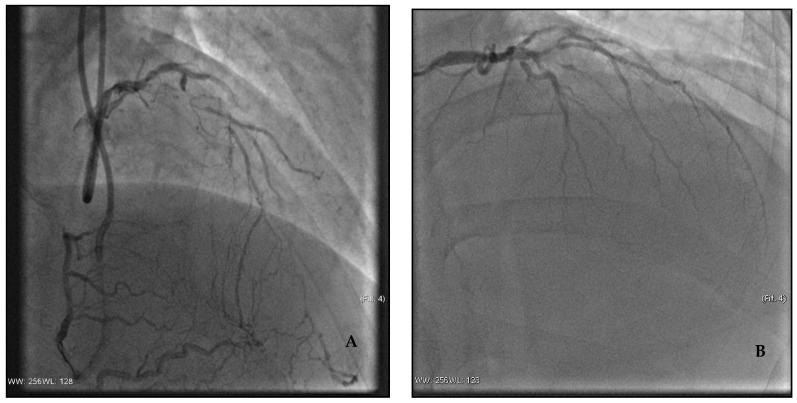
Coronary angiography. (**A**) Case 1: chronic total occlusion of the left anterior descending artery; (**B**) Case 2: chronic total occlusion of the left anterior descending artery in its middle tract and critical ostial stenosis of the first diagonal branch.

**Figure 3 jcm-11-01395-f003:**
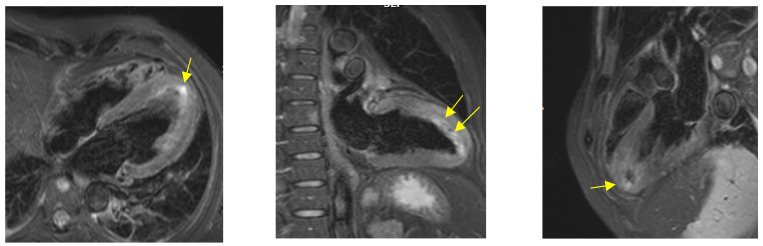
Case 2: cardiac magnetic resonance: T2w-STIR. Arrows indicate the presence of edema.

**Figure 4 jcm-11-01395-f004:**
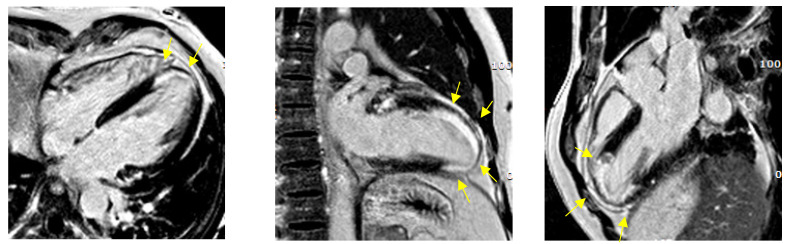
Case 2: cardiac magnetic resonance: arrows indicate late gadolinium enhancement (LGE).

**Table 1 jcm-11-01395-t001:** Randomized clinical trials comparing early and delayed CAG in patients with OHCA without STE on post-resuscitation ECG.

Trial	Patients1	Primary Outcome	Results
COACT (Coronary angiography after cardiac arrest without STE) [1] NTR4973	552	Survival at 90 days and one year	No difference
TOMAHAWK (Angiography after Out-of-Hospital Cardiac Arrest without STE) [22] NCT02750462	554	Survival at 30 days	No difference
DISCO (Direct or subacute coronary angiography in OHCA) [31] NCT02309151	1006	Survival at 30 days	Ongoing PILOT (79 patients): no difference
EMERGE (Emergency versus delayed coronary angiogram in survivors of OHCA) [29] NCT02876458	970	Survival at 180 days and neurological outcome	Ongoing
ARREST (Randomized trial of expedited transfer to a cardiac arrest center for non-STE OHCA) [30] NCT03872960	860	Survival at 30 days	Ongoing PILOT (63 patients): no difference
COUPE (Coronarography in OHCA) [33] NCT02641626	166	In-hospital survival and neurological outcome	Ongoing
PEARL (Early coronary angiography versus delayed coronary angiography) [32] NCT02387398	99	Survival at 180 days	Prematurely terminated (underpowered): no difference

CAG: coronary angiography, OHCA: Out-of-hospital cardiac arrest, ECG: electrocardiogram.

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
