# Peer review of "Percutaneous Coronary Revascularization after Out-of-Hospital Cardiac Arrest: A Review of the Literature and a Case Series"

_jcm, 2022, doi:10.3390/jcm11051395_

Round 1
Reviewer 1 Report
I congratulate the authors on their interesting review work. The goal of the work was clearly formulated and achieved. The use of clinical case reports as a background for the presented secondary data makes the text read with interest.
Author Response
We would like to thank again the reviewer for the kind comment and feedback provided.
Reviewer 2 Report
This is a case-based description of the patients with the out-hospital cardiac arrest, with a primary diagnosis of non-ST elevation myocardial infarction (NSTEMI), followed by an interesting discussion with presentation of various management protocols and the ambiguity of trials results.
According to available literature, patients after OHCA have a poor prognosis with high mortality rate at short- and 1 year follow-up period. Authors should include a 1-year outcome of the presented patients.
Furthermore, as Authors evidently wanted to include both case report plus the review of literature, I think that they should mention different management attitudes in patients undergoing cardiac arrest, such as therapeutic hypothermia (there are many studies not only with STEMI, but also NSTEMI patients, or mixed groups, e.g. Podolec, J., TrÄ…bka-Zawicki, A., Badacz, R., SiedliÅ„ski, M., Tomala, M., & BartuÅ›, K. et al. (2019). Chemokine RANTES and IL-1β in mild therapeutic hypothermia-treated patients after out-of-hospital sudden cardiac arrest. Advances in Interventional Cardiology/PostÄ™py w Kardiologii Interwencyjnej, 15(1), 98-106. https://doi.org/10.5114/aic.2019.83653), the urgent (immediate) or delayed angiography dependent on the patient's post-resuscitation neurologic status, the mater of a goal temperature (33°C vs <37.5°C), etc. Authors might also address the problem of cytokine storm having impact on the post-discharge poor survival prognosis in this subset of patients.
Although the Authors mentioned the key studies like TOMAHAWK and COACT studies, this issue should be more developed. Both studies showed no significant differences in clinical outcome among patients with out-of-hospital cardiac arrest between immediate and delayed coronary angiography at 90 days and at 1 year. Giving the numbers and proportion of patients suffering from adverse events would be interesting to the Readers, as the adverse outcomes are highly prevalent.
Author Response
Comment 1: According to available literature, patients after OHCA have a poor prognosis with high mortality rate at short- and 1 year follow-up period. Authors should include a 1-year outcome of the presented patients.
Response: We have included the 1-year follow up for both patients.
Comment 2: Furthermore, as Authors evidently wanted to include both case report plus the review of literature, I think that they should mention different management attitudes in patients undergoing cardiac arrest, such as therapeutic hypothermia (there are many studies not only with STEMI, but also NSTEMI patients, or mixed groups, e.g. Podolec, J., TrÄ…bka-Zawicki, A., Badacz, R., SiedliÅ„ski, M., Tomala, M., & BartuÅ›, K. et al. (2019). Chemokine RANTES and IL-1β in mild therapeutic hypothermia-treated patients after out-of-hospital sudden cardiac arrest. Advances in Interventional Cardiology/PostÄ™py w Kardiologii Interwencyjnej, 15(1), 98-106. https://doi.org/10.5114/aic.2019.83653), the urgent (immediate) or delayed angiography dependent on the patient's post-resuscitation neurologic status, the mater of a goal temperature (33°C vs <37.5°C), etc. Authors might also address the problem of cytokine storm having impact on the post-discharge poor survival prognosis in this subset of patients.
Response: We have added a review on the role of therapeutic hypothermia and the impact of cytokine storm, including the suggested reference in the manuscript.
Comment 3: Although the Authors mentioned the key studies like TOMAHAWK and COACT studies, this issue should be more developed. Both studies showed no significant differences in clinical outcome among patients with out-of-hospital cardiac arrest between immediate and delayed coronary angiography at 90 days and at 1 year. Giving the numbers and proportion of patients suffering from adverse events would be interesting to the Readers, as the adverse outcomes are highly prevalent.
Response: We thank the reviewer for pointing this out. We have incorporated the suggestions in the manuscript.
Reviewer 3 Report
I had the opportunity to review the manuscript by Scavelli et al. "Percutaneous coronary revascularization after out-of-hospital cardiac arrest: a review of the literature and case series" that has been submitted to the Journal for publication.
In this manuscript, the authors present a review on the indication of emergent coronary angiogram and percutaneous coronary intervention (PCI) in patients after out-of-hospital cardiac arrest (OHCA) when the ECG shows no ST-segment elevation. To date, randomized experiences showed no benefit of systematic emergent coronary angiogram and revascularization in this subset of OHCA. The authors describe two clinical cases whether emergent coronary angiogram with and without PCI was performed.
The manuscript is generally well-written, but some issues rase during revision:
1- The authors should include some more information on case 1 and case 2. I would recommend to include some figures summarizing ECG after OHCA and including also still images of the emergent coronary angiogram. For case 2 it would be interesting to include an image of cardiac magnetic resonance depicting edema and gadolinium enhancement.
2- In case 1 (page 3, line 144) it is not clear if the angiogram was emergent or performed thereafter.
3- It has to be clear that in this scenario, information regarding cardiac biomarkers as troponin is not available at the time you have to decide if the patient has an indication for emergent coronary angiogram. This point needs to be better addressed by the authors.
4- In table 1, the authors should add: 1) references to those published studies and 2) information regarding NCT numbers of those studies registered at clinical trials. This would help the readers in case they want further information on individual studies.
5- In case 1, the decision to perform PCI is questionable. As the patient reported no previous cardiac symptoms and acute coronary syndrome was properly excluded, current recommendations for PCI do not usually include asymptomatic patients. Mildly reduced ejection fraction described in the echocardiogram post-OHCA does not properly reflect ejection fraction after the acute event. Was inducible ischemia test performed before PCI? Was PCI in this case guided by coronary physiology? This point needs to be clarified.
6- In case 2, the indication of emergent PCI of a chronic total occlusion is even more questionable. The authors describe findings on cardiac magnetic resonance before discharge, with late gadolinium enhancement and signs of edema on the anterior wall. However, some of these findings could be related to the PCI itself. Please, describe the evolution of troponin levels on presentation and after PCI that could rule-out the presence of PCI-related myocardial infarction.
Author Response
Comment 1: The authors should include some more information on case 1 and case 2. I would recommend to include some figures summarizing ECG after OHCA and including also still images of the emergent coronary angiogram. For case 2 it would be interesting to include an image of cardiac magnetic resonance depicting edema and gadolinium enhancement.
Response: We agree that images would be interesting for the readers and therefore we included all the suggested figures.
Comment 2: In case 1 (page 3, line 144) it is not clear if the angiogram was emergent or performed thereafter.
Response: We have modified the text underling that the diagnostic angiogram was done in emergency however, since a culprit lesion was not identified, revascularization was not immediately performed.
Comment 3: It has to be clear that in this scenario, information regarding cardiac biomarkers as troponin is not available at the time you have to decide if the patient has an indication for emergent coronary angiogram. This point needs to be better addressed by the authors.
Response: This is absolutely correct. In the description of the cases we have moved laboratory test forward in the text to provide a temporal trend of the available information and diagnostic results. Moreover, we have clearly stated in the discussion that cardiac biomarkers were not available to help in the decision of performing emergent angiogram.
Comment 4: In table 1, the authors should add: 1) references to those published studies and 2) information regarding NCT numbers of those studies registered at clinical trials. This would help the readers in case they want further information on individual studies.
Response: We have modified the table accordingly.
Comment 5: In case 1, the decision to perform PCI is questionable. As the patient reported no previous cardiac symptoms and acute coronary syndrome was properly excluded, current recommendations for PCI do not usually include asymptomatic patients. Mildly reduced ejection fraction described in the echocardiogram post-OHCA does not properly reflect ejection fraction after the acute event. Was inducible ischemia test performed before PCI? Was PCI in this case guided by coronary physiology? This point needs to be clarified.
Response: Inducible ischemia testing was not performed prior to PCI. The decision to perform PCI was made in order to reduce the risk of sudden cardiac death and ventricular arrhythmias as we stated in the discussion
Comment 6: In case 2, the indication of emergent PCI of a chronic total occlusion is even more questionable. The authors describe findings on cardiac magnetic resonance before discharge, with late gadolinium enhancement and signs of edema on the anterior wall. However, some of these findings could be related to the PCI itself. Please, describe the evolution of troponin levels on presentation and after PCI that could rule-out the presence of PCI-related myocardial infarction.
Response: We modified the text underling that no significant increase in hs-Tnt was observed after the procedure. That rule-out the presence of PCI-related myocardial infarction.
Round 2
Reviewer 2 Report
I would like to congratulate Authors on interesting study. The Authors answered all issues and I do not have further comments.
Author Response
We thank this reviewers for the time and effort put into the review process, through which our paper was greatly improved.
Reviewer 3 Report
Most of my comments have been properly addressed by the authors.
I have only one more minor suggestions:
- In Figures 3 and 4, please add some markers (arrows, asterisks) to delimit the findings of edema (Figure 3) and late gadolinium enhancement (Figure 4). This would help readers not familiar with cMR to better understand the findings.
Author Response
We modified our Manuscript according to this suggestion.
We thank this reviewers for the time and effort put into the review process, through which our paper was greatly improved.